# Natural variations in maternal behavior shape anxiety and hippocampal neurogenesis in offspring

Fabio Grieco⬤, Pauline van Gelderen, Maryam Ali, Thomas Larrieu*⬤, Nicolas Toni*⬤

**Understanding the mechanisms of individualization of emotional traits is crucial for the development of personalized medicine in mood disorders. Although previous research focused on genetic differences or experimental manipulations of maternal care, it is unclear whether natural variations in maternal care in the absence of genetic variability influence anxiety-related behavior and neurogenesis. We addressed this question by observing inbred mouse maternal care and offspring in a longitudinal manner. We found that mothers spontaneously displaying low maternal care showed lower adult neurogenesis in both the olfactory bulb and the hippocampus compared with high maternal care mothers. In turn, low maternal care–reared pups exhibited increased anxiety as compared to high maternal care–reared pups, which was associated with decreased postnatal neurogenesis after weaning. These results highlight that in the absence of genetic diversity, natural variations in early-life experience are sufficient to shape brain plasticity and anxiety-related behavior in offspring. This underscores the relevance of maternal care and hippocampal neurogenesis in shaping personality-like traits related to mood disorders and inducing emotional individuality in early stages of life.**

## Introduction

In mammals, maternal behavior, usually emerging at or close to parturition, is a cornerstone of both physical and brain development, profoundly influencing the well-being of offspring (1). Human studies have highlighted the critical role of early parental care and bonding on a child's emotional development and cognitive functioning: for instance, children who experience consistent and nurturing maternal care tend to develop secure attachments, exhibit lower levels of stress, and perform better in cognitive tasks (2, 3, 4). Conversely, adverse childhood experiences, including neglect, are linked to higher risks of developing neuropsychiatric disorders (5) and cognitive impairment (6) later in life. In rodents, several studies have shown that artificially manipulating maternal care through different maternal deprivation and separation models can lead to diverse, profound, and long-lasting behavioral and physiological impairment in the offspring (reviewed in reference 1, 7). Indeed, maternal deprivation affects the offspring's hypothalamic–pituitary–adrenal (HPA) axis and neuroendocrine system, leading to brain remodeling and predisposition to anxiety- and depression-related behavior, and stress susceptibility (7). Inversely, adult offspring of mothers engaging in high maternal care (HMC) during the first postnatal week show less anxiety-like behavior and reduced corticotropin-releasing hormone mRNA expression and enhanced glucocorticoid negative feedback (8), along with stress responses (9, 10, 11). Adult rats that experienced maternal deprivation during postnatal development exhibit reduced expression of brain-derived neurotrophic factor and NMDA receptors in the hippocampus (12). In addition to inducing higher anxiety later in life, maternal separation also increases aggression in adult female dams, whereas it has the opposite effect in males (13). These emotional trajectories indicate a long-term influence of early-life experiences such as maternal care on offspring behavior.

However, these studies rely on artificial manipulations of maternal care, which do not reflect the subtle, naturally occurring variations in maternal behavior seen in rodents. Furthermore, it remains unclear whether natural variations in maternal care persist in the absence of genetic variability and whether these variations are sufficient to affect offspring brain development and behavior.

Despite sharing the same genetic background and controlled environmental conditions typically used in laboratory settings, adult C57BL/6J inbred mice exhibit remarkable interindividual differences in anxiety levels and hippocampal neurogenesis (14 Preprint, 15), which can lead to differences in stress vulnerability and in dominance behavior (15, 16, 17). These natural variations in adult individuals raise important questions about the mechanisms underlying the generation of individuality during development and suggest that nongenetic factors such as early-life experiences play a significant role in shaping individual traits in inbred mice. Here, we hypothesized that natural variations in maternal care contribute to the individualization offspring

Center for Psychiatric Neuroscience, Department of Psychiatry, Lausanne University Hospital, University of Lausanne, Prilly, Switzerland

Correspondence: Fabio.grieco@unil.ch; thomas.larrieu@chuv.ch; nicolas.toni@unil.ch
*Thomas Larrieu and Nicolas Toni contributed equally to this work

behavior. To test this possibility, we assessed the maternal behavior of C57BL/6J inbred mice and the natural variations in hippocampal development, neurogenesis, and anxiety in offspring.

# Results

## Inbred C57BL/6J mothers with spontaneously low maternal care (LMC) display less neurogenesis in the olfactory bulb and hippocampus

Rodents exhibit natural variations in maternal care. As to whether these interindividual differences are associated with differences in the levels of adult neurogenesis is not known. We observed and scored the behavior of dams with their pups during the first 2 wk post-parturition, monitoring individual variations in maternal parameters including arched-back nursing (ABN) and licking–grooming (LG) of pups (Figs 1A and S1A) (9). In addition to scoring these standard parameters, we also used occurrence in the nest (ON) as a simple method to assess maternal care. By segregating mice into two groups based on the mean total ON, we identified two distinct maternal profiles: HMC and LMC (Fig 1B). Pup retrieval test and ABN/LG behaviors are well-established gold standard to assess maternal care (18, 19, 20). Importantly, our analysis revealed a strong correlation between the percentage of ON and established maternal behaviors such as ABN and LG (Fig 1C; $R^2$ = 0.842; $P$ < 0.001). We further validated this method by showing that mothers exhibiting low levels of maternal care demonstrated significantly higher latency in retrieving pups within the home cage at postnatal day 5 (PND5), indicating reduced maternal responsiveness to separation (Fig 1D). This LMC profile was also evidenced by a significant reduction in the proportion of pups with visible milk spots at PND5 and PND6, highlighting compromised nursing behavior (Fig 1E). Finally, a principal component analysis (PCA) revealed a clear separation between the LMC and HMC mothers, with no overlap in behavioral patterns, supporting the validity of our classification system using ON (Fig 1F and G). This finding indicates that ON is a reliable parameter to simplify the evaluation of maternal behavior. Indeed, assessing the time spent in the nest offers a more straightforward, reliable, and reproducible measure of "maternal behavior" compared with individually monitoring passive nursing, arched-back nursing, pup contact, and licking–grooming, as it integrates all these behaviors, which predominantly occur within the nest environment. Interestingly, when ABN/LG frequency was normalized by the time mothers spent in the nest, it showed a negative correlation with total nest occurrence (Fig S1B), suggesting that high ON involved increased diversity of maternal behavior. Importantly, neither prenatal preparatory nesting behavior (Fig 1H) nor baseline anxiety levels (Fig 1I) of the mothers predicted maternal care quality, indicating these factors might be independent of postpartum maternal behavior.

A hallmark of mood-related behavior is the modulation of adult neurogenesis (AN), which primarily occurs in the subventricular zone (SVZ)–olfactory bulb (OB) and the dentate gyrus (DG) of the hippocampus. In the SVZ, newly generated neurons migrate to the

olfactory bulb, where they mature and enhance olfactory function crucial for odor detection and mother–pup interactions (21, 22, 23, 24). In the DG, adult-born neurons play an important role in buffering the stress response (25, 26, 27, 28, 29). These neurons are highly plastic and are themselves shaped by various environmental factors such as physical activity, aging, nutrition, disease, and stress (30, 31). Thus, adult neurogenesis is both a mediator of behavioral adaptation and a target of environmental and behavioral changes. LMC induced by experimental maternal separation showed that prolonged separation inhibits AN and emotional behavior in adult offspring (32, 33, 34, 35). We therefore examined neurogenesis in mothers and their offspring. LMC mothers displayed reduced adult neurogenesis in the DG and OB, as revealed by a reduction in the number of cells expressing the immature neuronal marker, doublecortin (DCX) (Fig 2A–D). A correlation matrix encompassing various behavioral and cellular parameters revealed that the latency to initiate pup retrieval negatively correlated with adult neurogenesis in the OB and DG of the mother, whereas ABN/LG positively correlated with adult neurogenesis (Fig 2E). In addition, the occurrence in the nest showed a positive correlation with adult neurogenesis in the OB and DG of the mother, suggesting that natural variations of adult neurogenesis in these brain regions under physiological conditions are intricately linked with interindividual differences in maternal behavior. Strikingly, this correlation analysis revealed a significant association between ON, pup retrieval, and ABN/LG, eating/drinking parameters, and litter size reflecting an important role in the number of pups per litter on the quality of maternal behavior (Figs 2E and S1C and D). Overall, our findings demonstrate that maternal behavior can be distinctly classified based on nest occurrence, with a "LMC mother" profile being closely associated with reduced adult neurogenesis in both the hippocampus and olfactory bulb.

## Natural variations in maternal care are associated with pups' USV emission and emotional individuality

Ultrasonic vocalizations (USVs) in pups are an early form of social communication reflecting emotional reactivity to environmental and social cues, such as maternal separation. Although often associated with distress- or anxiety-related responses—and reduced by anxiolytic agents (36, 37, 38)—USVs are more broadly considered indicators of emotional and motivational states. Here, we examined how natural variations in maternal behavior relate to this emotionally relevant communicative behavior in offspring. USV emission peaks during the first postnatal week (39, 40), offering an optimal window to study how early caregiving environments shape its development. Hence, we measured USVs at PND5 and PND9 to observe developmental changes in vocalization as the pups matured (Fig 3A). When separated from their mothers, LMC-reared male and female pups exhibited a heightened frequency (Fig 3B) and duration (Fig 3C) of vocalizations, compared with pups raised by HMC mothers, independently from the presence or absence of a milk spot at the time of assessment (Fig S2F and G). In pup mice, USVs can be divided into simple USVs, which are characterized by single, short-duration, frequency-

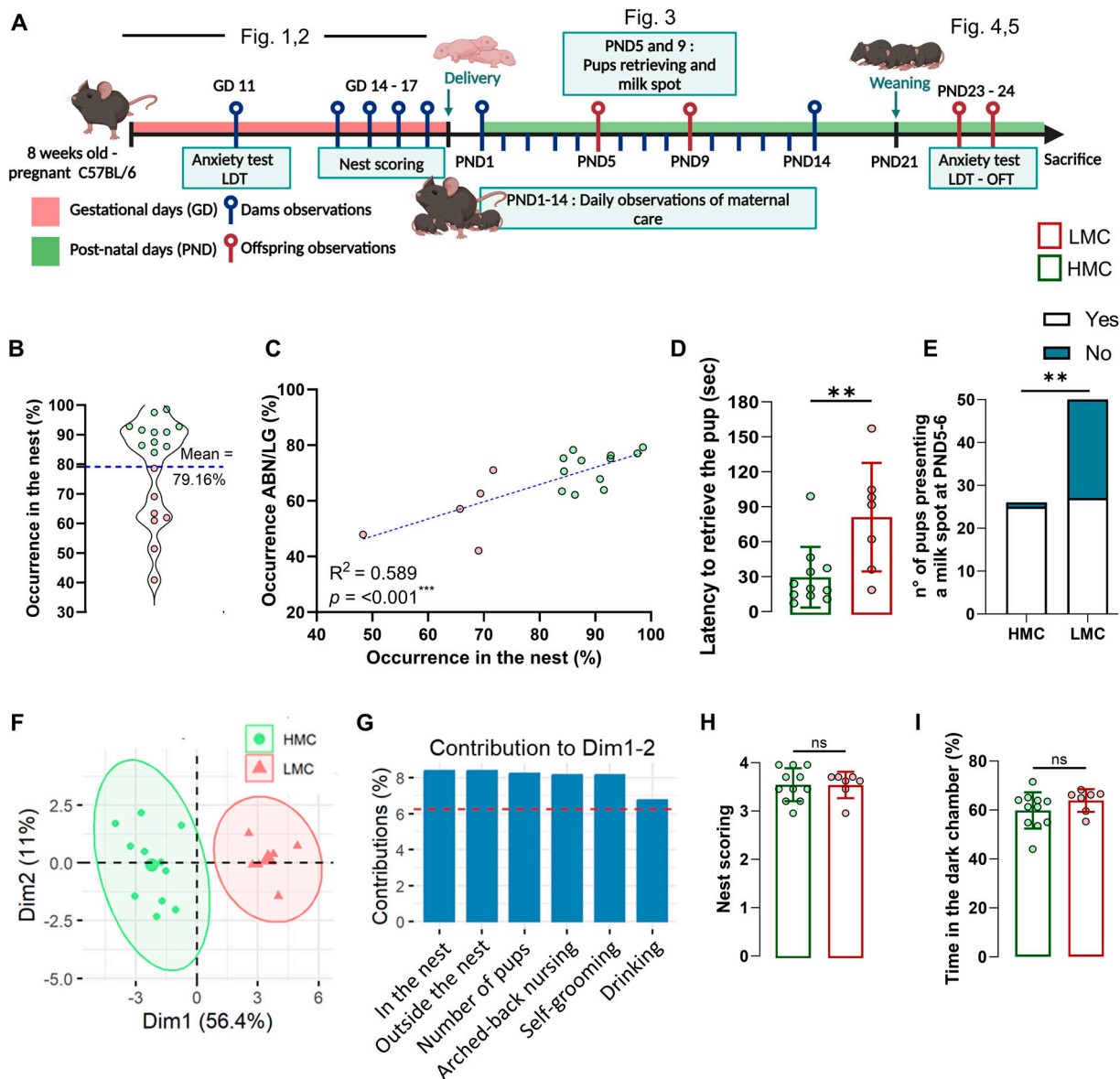

**Figure 1. Occurrence in the nest assesses natural variations of maternal care.**
**(A)** Schematic illustration of the experimental design. **(B)** Violin plot showing the distribution and the mean of occurrence in the nest used to discriminate low (LMC) from high maternal care mothers. **(C)** Correlation of ABN/LG occurrence with occurrence in the nest percentage (simple linear regression, $R^2 = 0.589$, $P = <0.001$). **(D)** Latency to retrieve pups at **PND5**. Each dot represents the **average time a dam took to retrieve her offspring** during the test. The mean latency per dam was used as one data point for statistical analysis ($t16 = 3.03$, $P = 0.008$, unpaired $t$ test, two-tailed, $n = 11$ and 7 per group). **(E)** Number of pups presenting a milk spot at PND5-6 (Fisher's exact test, two-sided, $P = <0.0001$). **(F)** Sample representation using the principal component analysis (PCA) ($n = 18$). **(G)** Relative contribution of the behavioral parameters to the dimensions 1 and 2 represented in the PCA. **(H)** Nest quality score assessed at gestational days 14 and 15 resulting from the ability to make a nest (Mann–Whitney test, $P = 0.993$, two-tailed, $n = 11$ and 7 per group). **(I)** Time spent in the dark chamber during a LDT at GD 11 ($t16 = 1.28$, $P = 0.219$, unpaired $t$ test, two-tailed, $n = 11$ and 7 per group). Histograms show average ± SEM, *$P < 0.05$, **$P < 0.01$, ***$P < 0.001$, ns, not significant.
Source data are available for this figure.

modulated calls and more elaborated calls such as "complex," "two-components," "composite," "harmonic," and "frequency steps" that exhibit multiple frequency modulations, varied pitch, and intricate temporal patterns (Fig S2D) (41). At PND5, we found that although LMC- and HMC-reared pups did not differ in terms of simple calls (Fig 3D), they did differ in terms of more elaborated calls reflected by an increased proportion of "complex,"

"composite," and "frequency steps" syntax in LMC-reared pups (Figs 3E and S2D). In addition, similar to a mouse model of autism (42), mean peak frequency was decreased (Fig 3F) and amplitude was increased (Fig 3G) in LMC-reared pups compared with HMC-reared pups. A similar but more pronounced pattern was observed at PND9 (Figs 3H–M and S2E), suggesting that the influence of maternal care on vocalization is established early in development

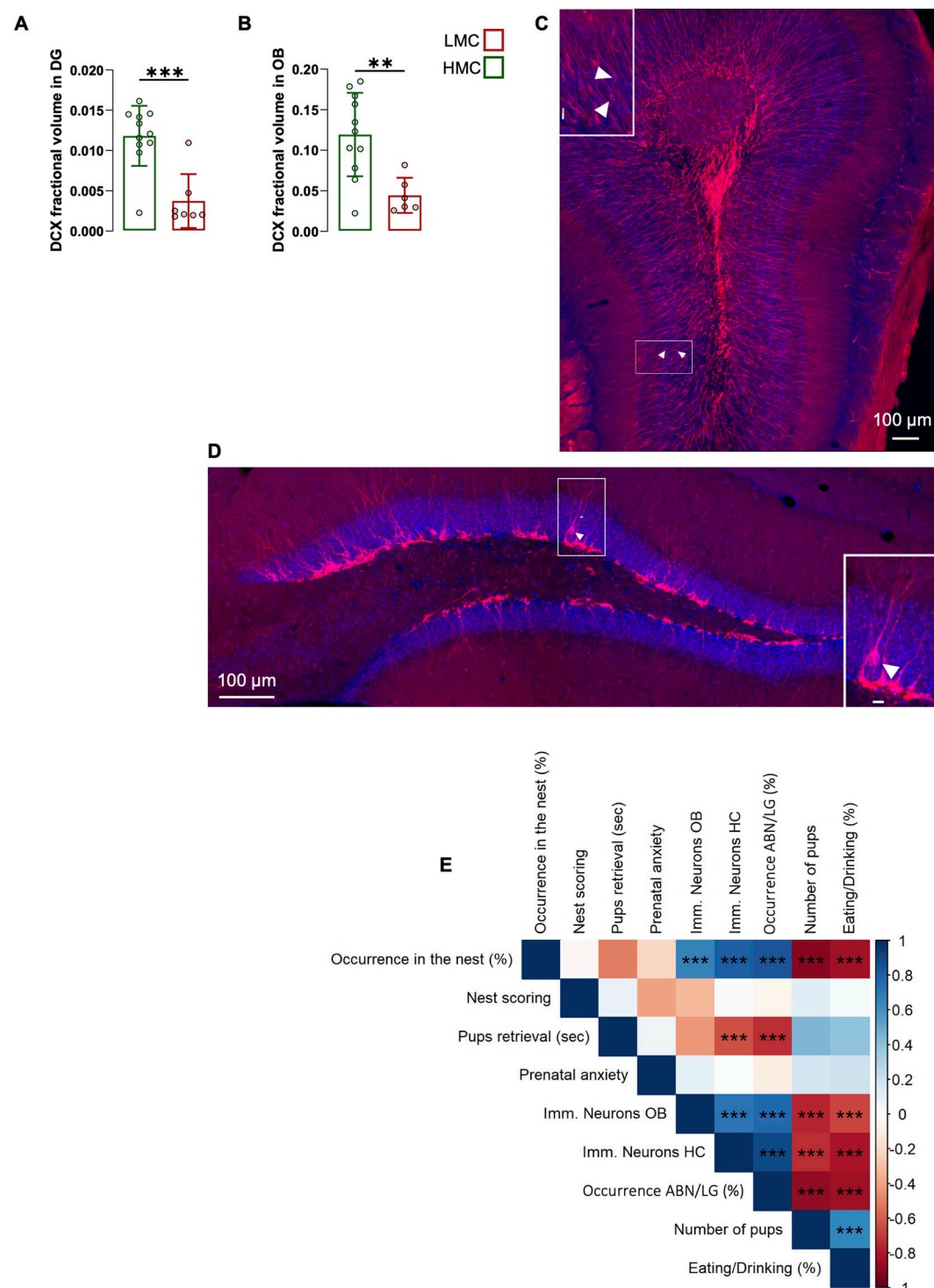

**Figure 2. Low maternal care mothers displayed decreased adult neurogenesis in the hippocampus and olfactory bulb.**

**(A)** DCX fractional volume in the dentate gyrus of low maternal care versus high maternal care (t16 = 4.66, $P$ = <0.001, unpaired $t$ test, two-tailed, n = 11 and 7 per group). **(B)** DCX fractional volume in the olfactory bulb (t15 = 3.37, $P$ = 0.004, unpaired $t$ test, two-tailed, n = 11 and 6 per group). **(C, D)** Confocal maximal projections of the dentate gyrus and olfactory bulb, immunostained for DCX (red) and DAPI (blue). Arrowheads point to DCX⁺ cells. Scale bars in insets: 100 μm. **(E)** Pearson's correlation matrix between maternal behavioral parameters, DCX fractional volume in the olfactory bulb (Imm.neurons OB) and hippocampus (Imm.neurons HC), prenatal anxiety, pup number, and eating/drinking. Correlation coefficients are shown together with significance levels. $P$-values were corrected for multiple comparisons using the Benjamini–Hochberg false discovery rate (FDR) procedure. Histograms show average ± SEM, *$P$ < 0.05, **$P$ < 0.01, ***$P$ < 0.001, ns, not significant.
Source data are available for this figure.

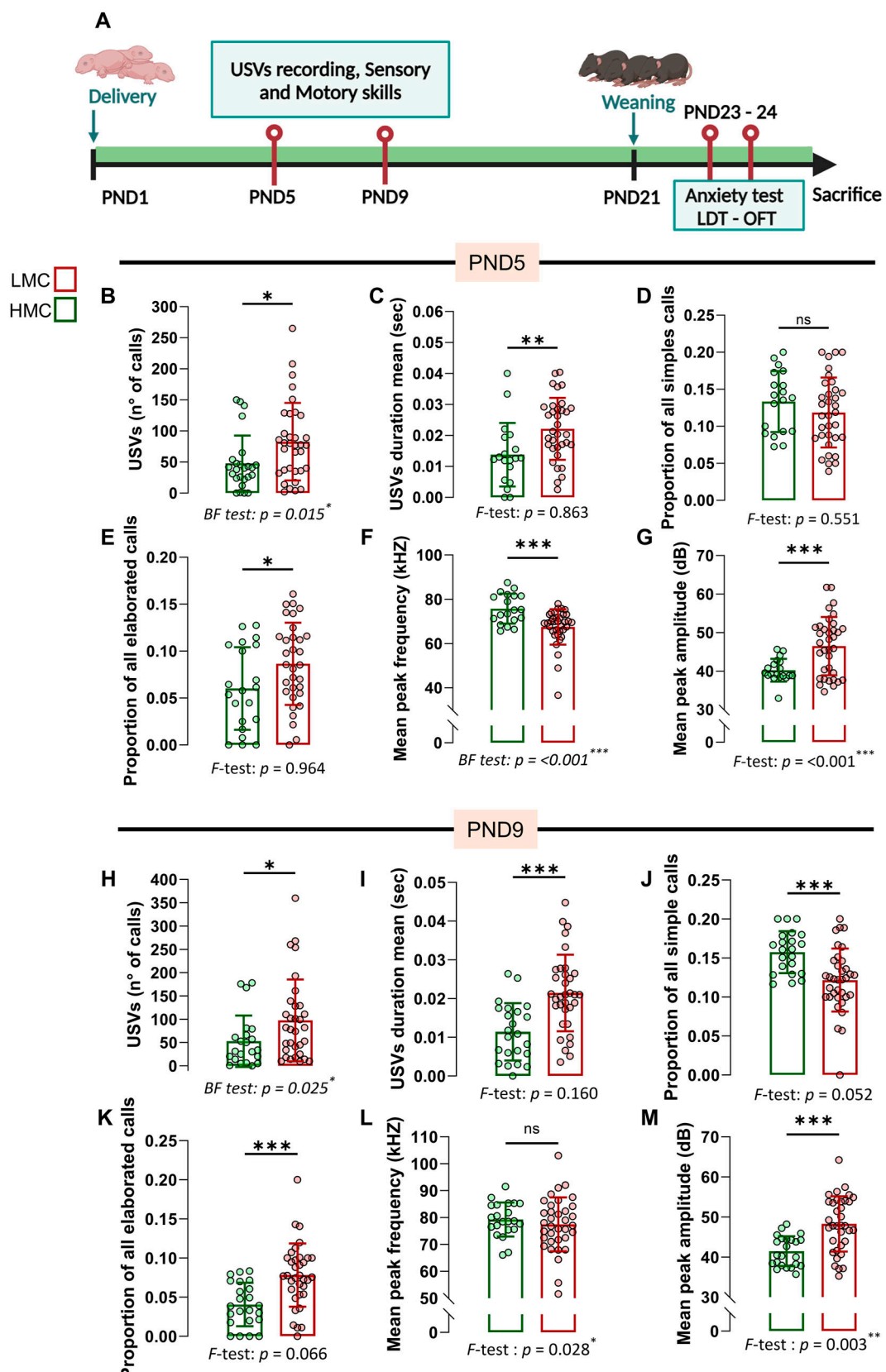

**Figure 3. Low maternal care–reared offspring displayed anxiety-related behavior assessed by USV recordings.**

(A) Experimental design for anxiety-related behavior assessment of offspring reared by low maternal care and high maternal care mothers of Fig 1. (B) Total numbers of calls upon acute maternal separation (Mann–Whitney test, $P = 0.021$, two-tailed, $n = 25$ and 34 per group). (C) Mean duration of the calls ($t51 = 2.89$, $P = 0.006$, unpaired

and persists as the pups mature. Finally, LMC-reared pups showed a significantly greater variance in most of the USV parameters we measured at both PND5 and PND9, suggesting that LMC is associated with the development of interindividual differences in isolation-induced vocalization emissions in pups at this early stage. To strengthen this observation with more sample size, we pooled data from two independent experiments at PND9 (Fig S3) and found that LMC-reared pups showed a significantly greater variance in all the USV parameters we measured except for the proportion of complex calls, confirming that natural variations of maternal care influence the development of emotional individuality in isolation-induced anxiety-related behavior in pups at early stage. Importantly, our observations at PND5 and PND9 revealed similar profile between LMC- and HMC-reared pups in terms of body weight, righting reflex, and cliff avoidance tests, suggesting that there is no developmental delay in LMC-reared pups as compared to HMC-reared pups (Fig S2A–C). Taken together, the analysis of USV indicates that LMC is associated with increased probability of vocalizations in both sexes at PND5 and PND9 during acute maternal separation and reveals the individualization of behavior.

### Natural variations in maternal care are associated with trait anxiety in juveniles

To investigate the potential influence of natural variations in maternal care on trait anxiety in offspring after weaning, the same mice were evaluated on a light–dark test (LDT) and an open field test (OFT), at PND22–23 (Figs 4A–C and S5A and B). We found that LMC-reared offspring exhibited increased anxiety-related behavior as compared to HMC-reared pups, revealed by increased time spent in the dark chamber of the LDT (Fig 4A) and in thigmotaxis during an OFT (Fig 4B). A composite anxiety score, calculated from normalized values of both tests, revealed higher anxiety levels in the LMC-reared group (Figs 4C and S6A). Importantly, there were no significant differences in the distance travelled between the groups in either the LDT or the OFT, and no interaction was found between maternal care and the sex of the pups (Fig S4A–E), indicating that variations in anxiety-related behavior were not confounded by differences in overall locomotor activity or sex. Finally, the groups did not differ in the variances in both the LDT and the OFT, suggesting that LMC did not trigger the development of interindividual differences in anxiety-related behavior at this age. Overall, these findings support the hypothesis that natural variations in maternal care quality have a significant impact on the development of trait anxiety in juvenile mice, highlighting the

critical role of early-life experiences in shaping emotional behavior.

### Natural variations of maternal care are associated with hippocampal neurogenesis in offspring

Natural variations in anxiety-related behavior under baseline conditions correlate with variations in adult neurogenesis (14 Preprint, 15). To investigate whether the observed behavioral variability in the offspring was associated with differences in neurogenesis in the DG of the hippocampus, we assessed cell proliferation by immunostaining against the cell proliferation marker Ki67. In addition, we identified immature neurons undergoing maturation by the presence of doublecortin (DCX). We found lower numbers of proliferating cells in the subgranular zone of the DG (Fig 5A and C) and lower DCX fractional volume (Figs 5B and D and S5C–F) in LMC- as compared to HMC-reared juvenile mice. LMC-reared pups displayed reduced AN in both males and females (Fig S4F and G). Correlation analyses further revealed that hippocampal neurogenesis in offspring negatively correlated with their anxiety score as previously observed in adult (Fig 5E) (14 Preprint, 15). Strikingly, occurrence in the nest during maternal care predicted the levels of hippocampal neurogenesis in juvenile offspring. We found no significant difference in the variance of markers of neurogenesis between LMC- and HMC-reared juveniles. Together, these findings indicate that maternal care is associated with hippocampal neurogenesis in weaned offspring, potentially underlying the observed behavioral differences in anxiety, and highlight the critical role of early-life experiences in shaping brain development and function in offspring.

### Natural variations of maternal care are associated with hippocampal neurogenesis at PND9

Building on our prior findings where LMC-reared pups exhibited increased USV calls at PND9 and a reduction in hippocampal neurogenesis at PND24, we sought to investigate whether these differences in neurogenesis could emerge at an earlier age, that is, PND9. To address this, we conducted a second longitudinal experiment where we first observed and scored maternal behavior from PND1 to PND9 as described above and segregated mothers into HMC and LMC groups based on the mean total ON (Figs 6 and S6). Importantly, the analysis of maternal care in experiment 1 revealed a strong correlation between the percentage of ON assessed during the first 2 wk post-parturition and the 1st wk (Fig S6A), indicating that 9 d of observation is sufficient to classify maternal care. Consistent with our initial observations,

---

*t* test, two-tailed, n = 19 and 34 per group). **(D)** Proportion of all simple calls (t51 = 1.14, *P* = 0.256, unpaired *t* test, two-tailed, n = 19 and 34 per group). **(E)** Proportion of all elaborated calls (t51 = 2.13, *P* = 0.038, unpaired *t* test, two-tailed, n = 21 and 32 per group). **(F)** Mean peak frequency (Mann–Whitney test, *P* = <0.001, two-tailed, n = 19 and 35 per group). **(G)** Mean peak amplitude (t48.62 = 4.32, *P* = <0.001, Welch's test, two-tailed, n = 19 and 35 per group). **(H)** Total number of calls (Mann–Whitney test, *P* = 0.033, two-tailed, n = 22 and 32 per group). **(I)** Mean duration of calls (t55 = 4.14, *P* = <0.001, unpaired *t* test, two-tailed, n = 23 and 34 per group). **(J)** Proportion of all simple calls (t55 = 3.67, *P* = <0.001, unpaired *t* test, two-tailed, n = 22 and 35 per group). **(K)** Proportion of all elaborated calls (t56 = 3.90, *P* = <0.001, unpaired *t* test, two-tailed, n = 23 and 35 per group). **(L)** Mean peak frequency (t54 = 0.84, *P* = 0.402, Welch's test, two-tailed, n = 22 and 35 per group). **(M)** Mean peak amplitude (t54 = 4.86, *P* = <0.001, Welch's test, two-tailed, n 23 and 35 per group). Histograms show average ± SD, *P < 0.05, **P < 0.01, ***P < 0.001, ns, not significant.
Source data are available for this figure.

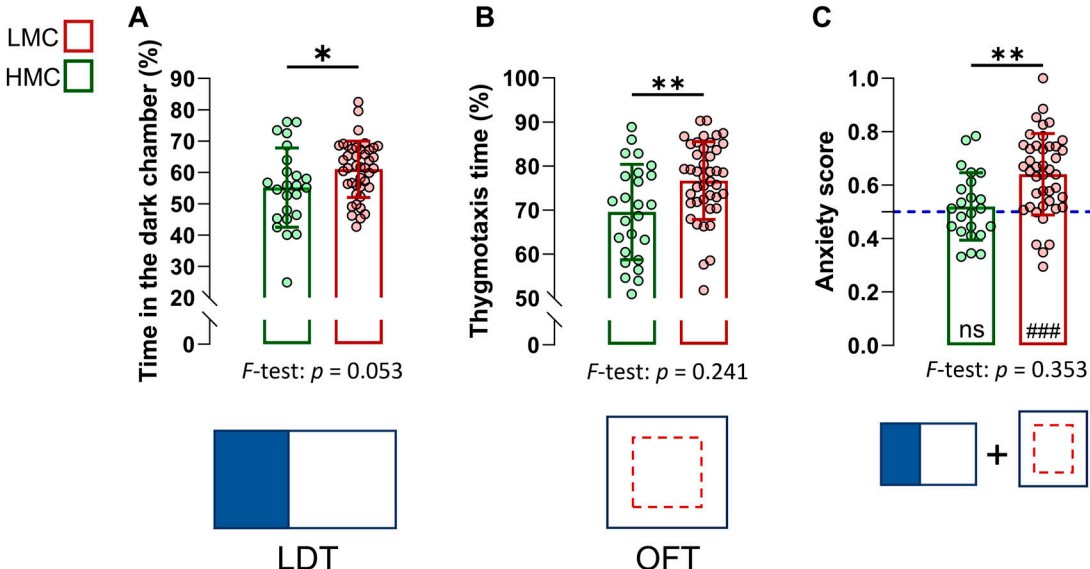

**Figure 4. Low maternal care–reared offspring displayed increased anxiety-related behavior at weaning.**
**(A)** Time spent in the dark chamber during a light–dark test at PND22 (t64 = 2.18, $P$ = 0.032, unpaired $t$ test, two-tailed, n = 24 and 42 per group). **(B)** Time spent in thigmotaxis during the open field at PND23 (t65 = 2.93, $P$ = 0.005, unpaired $t$ test, two-tailed, n = 25 and 42 per group). **(C)** Anxiety score derived from normalized data from LDT and OFT (t54 = 3.22, $P$ = 0.002, unpaired $t$ test, two-tailed, n = 23 and 42; t41 = 5.99, $P$ = <0.0001, one-sample $t$ test, two-tailed, n = 42). Histograms show average ± SD, *$P$ < 0.05, **$P$ < 0.01, ***$P$ < 0.001, ns, not significant. Comparison between the group mean and the hypothetical value of 0.5, to assess anxiety within each group, is shown within each histogram bar. One-sample $t$ test: #$P$ < 0.05, ##$P$ < 0.01, ###$P$ < 0.001, ns, not significant.
Source data are available for this figure.

this second cohort of mice reproduced the increased USV phenotypes in LMC-reared pups at PND9 (Fig 6A–G). Most importantly, LMC-reared pups displayed a significant decrease in the number of proliferating cells in the subgranular zone of the DG (Fig 6H and J), HopX-expressing stem cells (Fig 6H and I), and proportion of HopX+ stem cell that proliferated (Fig 6K) as compared to HMC-reared pups. At this developmental stage, we found statistical differences in the variances of Ki67-expressing cells between groups (Fig 6J). These findings indicate that naturally occurring LMC is associated with reduced hippocampal neurogenesis at early stage of postnatal development of offspring, potentially underlying the observed behavioral differences with moderate impact on individualization in the HMC-reared pups. In addition, correlation analyses further revealed the close relationship between proliferating cells in the dentate gyrus of PND9 pups and USV parameters (Fig 6L). Finally, ON parameter during maternal care observations predicts the levels of neural progenitor cell proliferation in the dentate gyrus of the hippocampus of PND9 pups (Fig 6L) revealing a crucial role of maternal care quality in brain plasticity in offspring at an early stage of development.

# Discussion

Here, we used inbred mice to assess the effect of natural variations of maternal care on the longitudinal development of offspring behavior and hippocampal neurogenesis in the absence of genetic differences. We found that natural variations in maternal care are associated with interindividual differences in adult neurogenesis in the OB and the DG of dams and with natural variations of anxiety

traits and hippocampal neurogenesis in their offspring during early developmental stages (PND9 and PND23).

Maternal behavior is not uniform even within inbred C57BL/6J strain, and mothers can be segregated into LMC and HMC, which is consistent with previous studies showing individual variations in maternal behavior after 2-wk monitoring of dams with their pups (9, 43). Here, we combined established gold standard parameters with a simple method that evaluates maternal care based on the occurrence in the nest (ON) (44). This approach revealed a strong correlation between ON and traditional maternal behavior such as arched-back nursing (ABN) and licking–grooming (LG) (9), validating ON as a reliable proxy for comprehensive maternal behavior assessment. Moreover, this method circumvents the need for separate monitoring of passive nursing, ABN, pup contact, and LG, thereby enhancing observational efficiency and consistency. Further validating the ON parameter, we found that LMC mothers exhibited significantly higher latencies in pup retrieval, indicative of reduced maternal responsiveness to pup separation as observed after gestational stress (18) and a reduction in the proportion of pups with visible milk spots at PND5 and PND6, indicating impaired nursing behavior (23).

As olfaction is essential to mouse maternal behavior (45), and motherhood is associated with increased neurogenesis in the OB (46, 47), we compared neurogenesis in the OB and the DG of LMC and HMC mothers. We found that LMC mothers, as compared to HMC mothers, exhibited reduced adult neurogenesis in both the DG and the OB, indicated by a reduced expression of the immature neuronal marker doublecortin (DCX). These results are consistent with previous studies reporting that the normal expression of postpartum maternal behavior requires increased

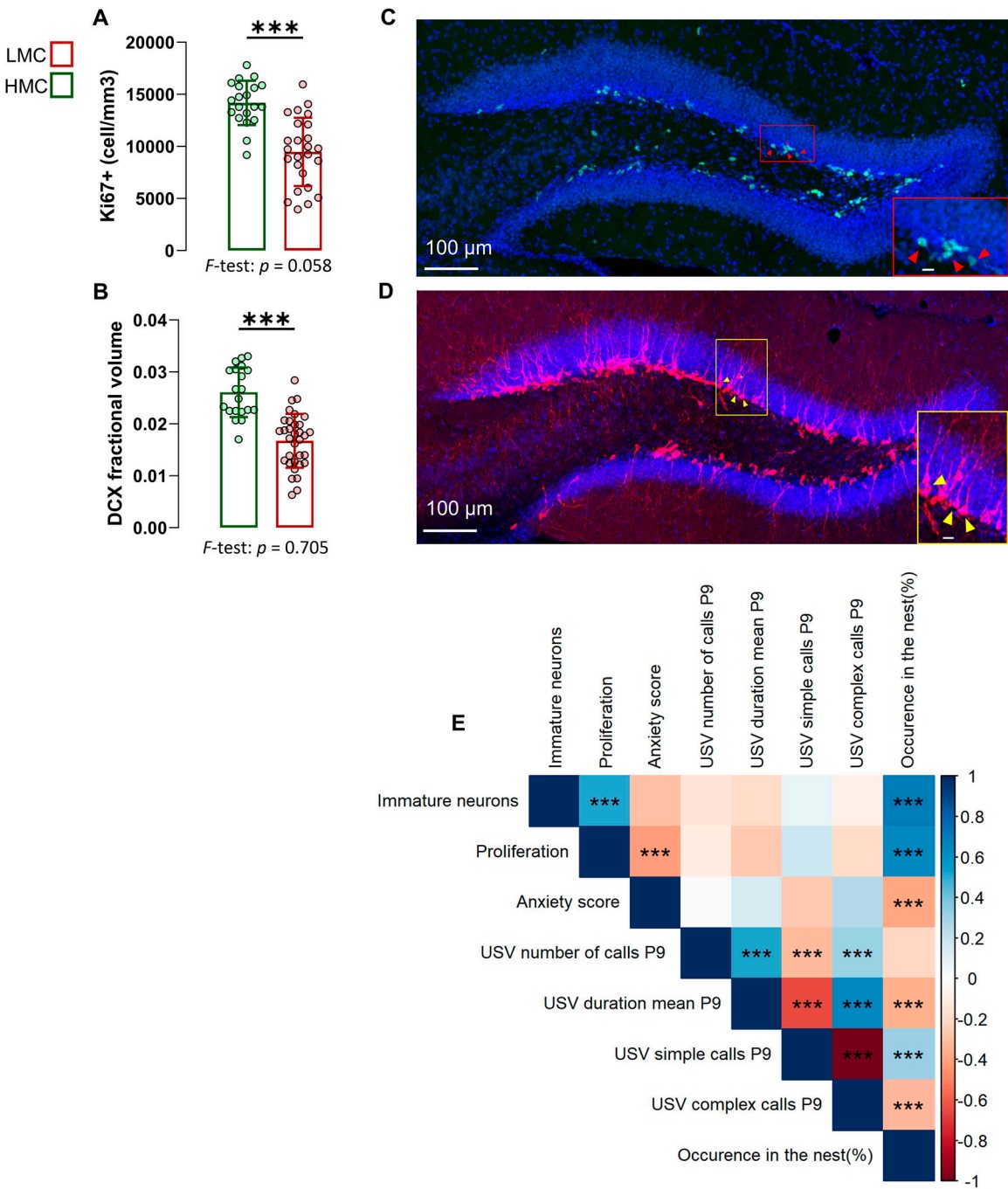

**Figure 5. Juvenile low maternal care (LMC)–reared offspring displayed decreased hippocampal neurogenesis.**
**(A)** Quantification of Ki67-positive cells in the dentate gyrus of the hippocampus of LMC- and high maternal care–reared offspring ($t43 = 5.55$, $P = <0.001$, unpaired $t$ test, two-tailed, n = 25 and 20 per group). **(B)** DCX fractional volume in the dentate gyrus of the hippocampus of LMC- and high maternal care–reared offspring ($t51 = 6.55$, $P = <0.001$, unpaired $t$ test, two-tailed, n = 20 and 33 per group). **(C, D)** Representative confocal microscopy image of the dentate gyrus immunostained for Ki67 ((C) red cells highlighted with yellow arrows), and DCX ((D) white cells highlighted with yellow arrows) and DAPI ((C, D) blue). **(E)** Pearson's correlation matrix between selected USV features, neural progenitor cell proliferation and immature neurons in the dentate gyrus, juvenile anxiety, and the time spent in the nest by the mother. Correlation coefficients are shown together with significance levels. *P*-values were corrected for multiple comparisons using the Benjamini–Hochberg false discovery rate (FDR) procedure. Histograms show average ± SD, *$P < 0.05$, **$P < 0.01$, ***$P < 0.001$, ns, not significant. Inset scale bars: 10 $\mu$m.
Source data are available for this figure.

adult neurogenesis in the SVZ of pregnant female mice (46, 47). However, nulliparous female rats that display maternal care upon pups' exposure show more proliferating cells in the SVZ compared with females that were never exposed to pups or females that did not display any maternal care (48). Together with these studies, our results suggest that although adult neurogenesis supports

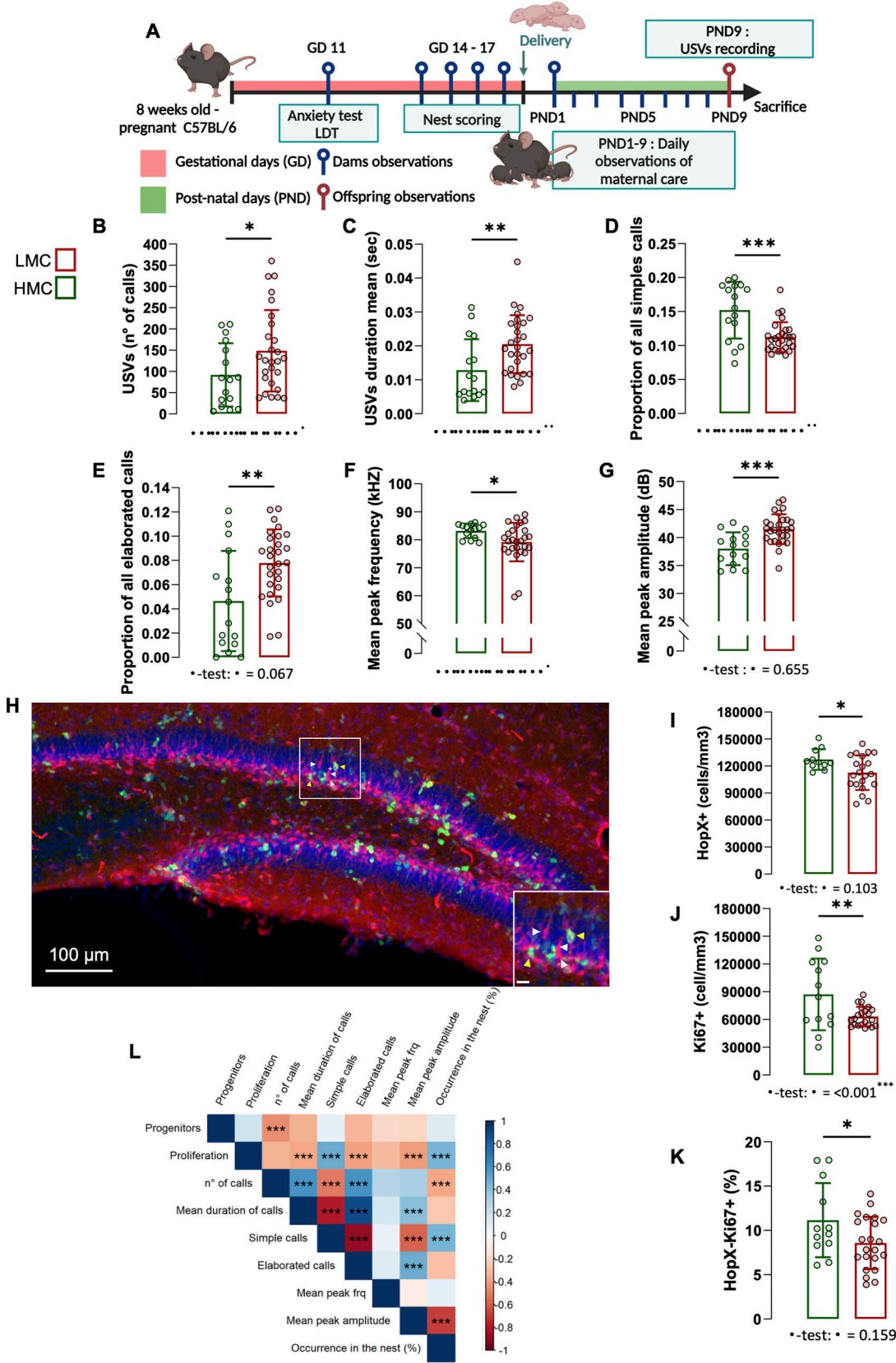

**Figure 6. LMC-reared offspring displayed decreased hippocampal neurogenesis at PND9.**
(A) Schematic representation of the experimental design. (B) Total numbers of calls upon acute maternal separation (Mann–Whitney test, *P* = 0.039, two-tailed, n = 16 and 27 per group). (C) Mean duration of the calls (Mann–Whitney test, *P* = 0.004, two-tailed, n = 16 and 28 per group). (D) Proportion of all simple calls (Mann–Whitney

maternal behavior in dams, the act of providing maternal care itself may also be an important modulator of neurogenesis, specifically targeting differentiating cells regardless of pregnancy or actual motherhood (49).

To investigate the relationship between interindividual differences in maternal care and pup anxiety-related behavior at an early stage of life, we measured USV emissions at PND5 and PND9. During the early postnatal period of rodent development, USVs constitute the primary acoustic signaling mechanism that serves as a crucial ethological biomarker to assess the emotional state of the pups (50). Transient maternal separation is a well-known paradigm for eliciting USVs in rodent offspring. This acute separation-induced USV increases during the 1st wk of pup life, reaching a peak around PND7, and decreasing until PND14 (40). Here, we observed that LMC was associated with higher USV emissions in pups, reflected by an increase in the number and duration of calls. The increase in vocalizations in LMC-reared pups suggests an increased anxious response to separation when compared to HMC-reared pups (51). This is supported by studies indicating increased levels of anxiety-related behavior in adult offspring mice raised by mothers displaying LMC (9). Further supporting this idea, anxiolytic compounds and selective serotonin reuptake inhibitors suppress separation-induced USV production in rat pups (52, 53). In addition to increased USV frequency, we found that reduced maternal care was also associated with more elaborated calls in pups. For instance, LMC-reared pups exhibited a decrease in the proportion of "short" and "down" calls referred to as "simple calls," whereas "complex," "two-components," and "composite" calls referred to as "elaborated calls" were increased. This observation suggests that maternal care may influence communication patterns during early postnatal development. The increased anxiety in LMC-reared pups was confirmed by observations after weaning. Indeed, at that stage offspring from LMC mothers exhibited heightened anxiety, as evidenced by an increased time spent in the dark chamber of a light–dark box and in thigmotaxis during an open field test. Importantly, this increased anxiety was not attributable to differences in general locomotor activity. Thus together, these results confirm that LMC rearing is associated with increased anxious behavior in a longitudinal manner.

Emotional individualization refers to the development of unique emotional traits and is shaped by a combination of genetic, environmental, and neurobiological factors. The role of maternal care in pup development has been studied using experimentally induced maternal separation (7, 8, 9, 10, 11, 12, 13, 32, 33, 34, 35, 54, 55) or genetically diverse lines of rodents (33, 34, 56, 57, 58, 59, 60, 61, 62, 63, 64). Here, we showed that in the absence of manipulations or genetic diversity, natural variations in maternal behavior are correlated with variability in pups' behavior and hippocampal neurogenesis. It has recently been shown that environmental enrichment, by exposing animals to "nonshared experience," induces significant variability in behavior in inbred C57BL/6J mice (65, 66). For pups at early developmental stages, variations in maternal care can be perceived as a naturalistic environmental enrichment, which contributes to their behavioral diversity. Furthermore, the higher behavioral variability among LMC-reared pups may result from the reduced presence in the nest of LMC mothers. This, in turn, may induce a higher explorative behavior from the pups out of the nest, leading to nonshared experience that may further contribute to increased behavioral variability induced by natural variations in maternal care. Thus, by increasing the diversity of rearing circumstances, natural variations of maternal care may represent the first steps of environmental enrichment in an individual's lifetime.

The source of natural variations in maternal care remains, however, unclear. The inverted correlation between litter size and occurrence in the nest (Fig S1C) supports the view that litter size decreases maternal care (67, 68). However, dams of similar litter size displayed variations in occurrence in the nest, suggesting that maternal care is also influenced by other factors in addition to litter size. In addition, it remains unclear whether the first steps of pup's individualization appear before the peak of maternal care (between PND5 and PND9) and whether they involve other mechanisms than hippocampal neurogenesis, such as epigenetic modifications (9, 69). Further investigation is required to shed light on these sources of variability.

Although we have not addressed it here, the interindividual variability in hippocampal neurogenesis and anxiety we observed in pups likely extends into adulthood and may contribute to the variability in trait anxiety and adult hippocampal neurogenesis (14 *Preprint*, 15, 27). Adult hippocampal neurogenesis inhibits anxious behavior, as demonstrated by the anxiety induced by the pharmacological (15, 70) or genetic inhibition (27) of adult neurogenesis. In turn, mood impairment induces alterations in adult neurogenesis (25, 26, 27, 28, 29, 30, 31). This feedback regulation may sustain anxiety throughout adulthood and underlie susceptibility to mood disorders, as well as cognitive impairment (71), dominance behavior (15), or maternal behavior.

Together, the results presented here indicate that despite strong genetic homogeneity and similar housing conditions, inbred laboratory mice display large differences in maternal behavior and adult neurogenesis, which are related to strong variability in anxiety and hippocampal neurogenesis in offspring. These mechanisms of behavioral and physiological individualization may increase group fitness in times of reduced genetic diversity and underlie individual susceptibilities to brain diseases.

---

test, $P$ = 0.003, two-tailed, n = 16 and 28 per group). **(E)** Proportion of all elaborated calls (t42 = 3.01, $P$ = 0.004, unpaired $t$ test, two-tailed, n = 16 and 28 per group). **(F)** Mean peak frequency (Mann–Whitney test, $P$ = 0.036, two-tailed, n = 14 and 27 per group). **(G)** Mean peak amplitude (t41 = 3.95, $P$ = <0.001, unpaired $t$ test, two-tailed, n = 15 and 28 per group). **(H)** Confocal maximal projection of the dentate gyrus immunostained for Ki67 (in red, highlighted with yellow arrows) and HopX (in green, highlighted with yellow arrows). **(I)** Quantification of HopX-positive cells (t30 = 2.31, $P$ = 0.027, unpaired $t$ test, two-tailed, n = 11 and 21 per group). **(J)** Quantification of Ki67-positive cells (t13.05 = 2.18, $P$ = 0.048, Welch's test, two-tailed, n = 13 and 22). **(K)** Percentage of HopX$^+$-Ki67$^+$ cells over the total amount of HopX$^+$ cells (t31 = 2.35, $P$ = 0.025, unpaired $t$ test, two-tailed, n = 12 and 21 per group). **(L)** Pearson's correlation matrix between selected USV features, cellular proliferation, and neuronal progenitors in the dentate gyrus. Correlation coefficients are shown together with significance levels. $P$-values were corrected for multiple comparisons using the Benjamini–Hochberg false discovery rate (FDR) procedure. Histograms show average ± SD, *$P$ < 0.05, **$P$ < 0.01, ***$P$ < 0.001, ns, not significant. Inset scale bars: 10 $\mu$m. Source data are available for this figure.

## Materials and Methods

### Animals

All experiments were performed with the approval of the Cantonal Veterinary Authorities (Vaud, Switzerland) and carried out in accordance with the European Communities Council Directive of 24 November 1986 (86/609EEC). All experiments were performed on C57BL/6J male and female mice obtained from Janvier Laboratories (France). After arrival at 7-wk-old, mice were left undisturbed for a 1-wk acclimation period. Twenty-five nulliparous females were used to form breeding pairs, separated from the male after 1 d of mating, and then single-housed for the rest of the experiment. Mice were weighed weekly to track their health status and pregnancy. Mice were randomly distributed into breeding pairs. Mice were maintained under standard housing conditions on corncob litter in a temperature (23°C ± 1°C)- and humidity (40%)-controlled animal room with a 12-h light–dark cycle (8h00–20h00), with unlimited access to food and water. All tests were conducted during the light period.

### Preparatory nesting behavior

Twenty-four hours before the nest scoring, the nest material was replaced with four fresh squares of cotton nestlet, and the mothers remained undisturbed until the next day when scoring was made. On gestational days 14, 15, and 16 at 10 am, nests were manually scored as previously described (6) using an adapted 5-point scale from 0 to 4: 0, nesting material is not shredded, no visible nest site; 1, nesting material is completely or partially shredded, identifiable nest but flat; 2, nesting material is completely or partially shredded, saucer-shaped nest; 3, nesting material is completely shredded, nest walls have enough height to cover the entire animal, but no closed roof; and 4, nesting material is completely shredded, nest covers the entire animal including a complete roof. When hesitating between two scores, 0.5 pts were used.

### Anxiety tests

To assess anxiety-related behavior, 67 pups and 18 dams were tested in the light–dark test (LDT) and open field test (OFT). All behavioral assessments were performed by a single experimenter blinded to the maternal care the pups received.

#### Light–dark test (LDT)

The apparatus used for the LDT consisted of a white wooden box with two compartments. One was a square compartment without a lid, serving as the light side (40 × 40 cm), whereas the other was a smaller rectangular compartment with a lid, creating the dark side (20 × 40 cm). These two compartments were connected by a 5 × 5 cm door, and the entire apparatus had a height of 30 cm. The center of the lit compartment maintained a stable luminosity of 400 lux, whereas the dark compartment remained without any light source. Mice were introduced into the apparatus in the light side, facing the door, and allowed to explore for a duration of 5 min. The mice's movements were tracked and recorded using ANY-maze software.

In this test, anxiety-like behavior was evaluated based on the time spent in the dark compartment.

#### Open field test (OFT)

The OFT was conducted as previously described (17) in a rectangular arena (50 × 50 × 40 cm³) illuminated with dimmed light (25 lux). Mice were introduced near the wall of the arena and allowed to explore for 10 min. Analyses were performed using ANY-maze tracking software by drawing a virtual zone (15 × 15 cm²) in the center of the arena defined as the anxiogenic area. Several parameters were analyzed, including the total distance travelled and the time spent in the different zones.

### Anxiety score

The anxiety scores encompassed several anxiety tests to get a general profile of anxiety as previously described (16, 17, 72, 73). The score is derived from the normalization of the values for the combination of individual anxiety tests (time spent in the dark chamber during LDT and time spent in thigmotaxis in an OFT). The normalization involved adjusting each animal's value by subtracting the minimum value of the entire population and then dividing this result by the difference between the maximum and minimum values of the entire population: (x – min value)/(max value – min value). This method generates scores distributed on a scale from 0 to 1, with a score of 1 indicating high anxiety.

### Maternal behavior monitoring

Starting on postnatal day (PND) 1, maternal behavior was systematically recorded twice daily (at 09:00 and 18:00) for 2 wk during the light phase, when nursing is most frequent. Behavioral monitoring was performed by a single experimenter who was blind to the identity of the mice. Using an adapted protocol (9), cages were observed for 5 s every 4 min, with each session comprising 10 snapshots per cage, leading to a total of 280 observations per dam over the experiment. Behaviors recorded for each mother were as follows: in the nest: passive nursing, arched-back nursing, contacting pups, licking, or grooming pups; manipulating nest bedding or out of the nest: eating, drinking, self-grooming, rearing. Licking–grooming and arched-back nursing are the typical maternal behaviors that serve as the principal criterion in literature to discriminate between high and LMC (9, 34, 43, 63). Hence, the combined percentage of licking–grooming and arched-back nursing (ABN/LG) occurrences in both the morning and evening was calculated and correlated to the occurrence in the nest. Dams were classified as high or LMC groups based on percentages of occurrence in the nest above or below the cohort mean, which remained consistent across experiments.

### Ultrasonic vocalization (USV) recording

USVs from 104 pups were recorded upon isolation from the home cage at PND5 and PND9. Recordings were conducted by a single experimenter blinded to the maternal care condition of the pups. Briefly, each individual pup was separated from its mother and littermates and was assessed for the presence of a milk spot. After

this, pups were placed in a phonically isolated chamber (63 × 38 × 42 cm) containing a microphone (UltraSoundGate 116H; Avisoft Bioacoustics) suspended from the top of the chamber ~10 cm from the bottom of the recording chamber. The pup was gently placed in a plastic cup within the recording chamber to minimize movement. Each pup was recorded for 3 min in a range between 10 and 120 kHz. Call detection was provided by an automatic threshold-based algorithm (amplitude threshold: –20 dB; hold time: 5 msec.). The accuracy of call detection was verified manually. USV calls were then analyzed using Avisoft-SASLab Pro software to determine the number of calls, the mean peak frequency, the mean peak amplitude and the mean duration of calls, simple USVs characterized by single, short-duration, frequency-modulated calls, and elaborated calls such as "complex," "two-components," "composite," "harmonic," and "frequency steps" that exhibit multiple frequency modulations, varied pitch, and intricate temporal patterns.

### Milk spot detection

To assess whether pups had been fed by their mothers, the presence of a visible milk spot in the abdomen was assessed (74). Owing to the natural skin transparency at this age, the bright spot of the milk-filled stomach could be readily detected. Between hours 9 and 11 of postnatal days (PND) 5 and 6, 76 pups were examined by a single-blinded experimenter, and the milk spot was scored in a binary manner, as either present or absent.

### Righting reflex test

To assess the development of the pups' motor ability, a righting reflex test was performed on 69 pups, at PND5 and PND9. The pup was held in a supine position 5 s by the experimenter, with all four paws upright. The pup was then released immediately. Righting occurs when the pup flipped over to prone position on the table. The time spent by the pup to flip was used to compute a score: 3, 0–10 s to flip; 2, 10–20 s to flip; 1, 20–30 s to flip; and 0, > 30 s to flip.

### Cliff avoidance test

The cliff avoidance test was performed on 69 pups as already described (75) to assess labyrinth reflexes, as well as the strength and coordination ability of the pups on PND5 and PND9. The pup was placed at the edge of a flat surface, such that the forepaws and snout of the pup were over the edge. The correct outcome is a protective response, where the pup turns away from the edge of the cliff. Behavior was scored as follows: 0, no movement or falling off the edge; 1, attempts to move away from the cliff but with hanging limbs; and 2, successful movement away from the cliff.

### Pups' retrieval test (PRT)

The pup retrieval test stands out as a predominant test in fundamental and preclinical rodent research for the assessment of maternal behavior (76). This test quantifies the mother's retrieval response to the removal of a pup from the nest. The test was conducted on PND5 and PND9 after 5 min of pup's separation during which USV recording, righting reflex, and cliff avoidance

tests were performed. Pup retrieval was defined as picking the pup up in the mouth and transporting into the nest. A trial started as the mother was in the nest, and a pup was placed in the most distant corner relative to the nest. The latency to retrieve the pup was measured using a stopwatch. If the pup was not retrieved within 180 s, it was placed back into the core of the nest.

Because of maternal habituation to pup retrieval, the test was performed using a maximum of four pups per litter, when at least four pups were available.

### Histology

After all the behavioral experiments, the mice were euthanized with a lethal injection of pentobarbital (10 ml/kg; Sigma-Aldrich) and transcardially perfused with saline solution (NaCl 0.9%) followed by 4% PFA solution (Sigma-Aldrich). Brains were removed and postfixed with PFA 4% at 4°C overnight. Then, brains were transferred in a 30% sucrose solution for 3 d before mounting in OCT compound and stored at –20°C until slicing. Coronal frozen sections of a thickness of 35 $\mu$m were sliced with a cryostat (CM3050S; Leica) to obtain hippocampal and OB sections conserved in a cryoprotectant solution (30% glycerol, 30% ethylene glycol, and 40% PBS 1 M) at –20°C until immunofluorescence staining.

### Immunohistochemistry

Brains from 18 dams and 87 pups, randomly and blindly selected across litters, were processed for the evaluation of neurogenesis. To avoid overrepresentation of individual litters, a comparable number of samples were blindly selected from each litter, irrespective of the maternal care condition received. One out of six slices containing hippocampal or olfactory bulb (OB) tissue was chosen to cover the whole dentate gyrus and OB and used for immunostaining. Slices were then incubated for 48 h at 4°C in PBST1% containing 5% of normal goat or donkey serum with the following primary antibodies: rabbit anti-DCX (1:500, 4604S; Cell Signaling Technology), rabbit anti-Ki67 (1:500, ab15580; Lucerna Chem), and mouse anti-Hopx (1:500, sc-398703; Santa Cruz). After 48 h of incubation, the sections were rinsed 3 × 10 min in PBST1% and incubated for 3 h in either of the following secondary antibodies in PBST1% and 5% of normal goat or donkey serum: goat anti-rabbit Alexa 594 (1:500, A11037; Life Technologies), goat anti-rabbit Alexa 488 (1:500, A11034; Invitrogen), donkey anti-rabbit (1:500, A31573; Invitrogen), and donkey anti-mouse Alexa 488 (1:250, A21202; Invitrogen). Sections were then rinsed 10 min in PBS1x and incubated in DAPI (1:1,000) to reveal nuclei before being rinsed again 3 × 10 min in PB 0.1 M. All images of the immunostained sections of OB and hippocampus were acquired with a Nikon NI-E spinning disk microscope with a 20X objective (20x/0.75; Nikon Plan APO). Nikon NIS-Elements imaging software was used to do 3D supervised automatic image analysis.

The number of labeled cells was counted for each section, in the entire thickness of the section. In the dentate gyrus, cells expressing Ki-67 and HopX were counted in an area containing the subgranular zone and the granule cell layer (GCL). The fractional volume occupied by DCX immunostaining was obtained

through supervised 3D image analysis of brain sections using a constant threshold applied across all slices. This parameter was calculated as the proportion of DCX-positive voxels over the total voxel number, analyzed throughout the full section thickness, and multiplied by the number of sections evaluated per animal. In the DG, the DCX fractional volume was measured in an area including the SGZ, GCL, and the initial portion of the molecular layer (ML). In the OB, it was measured within the GCL. As immature neurons express DCX in the soma and the proximal dendrites, changes in the fractional volume can be influenced by changes in either the number of DCX-expressing cells, the dendritic length of each cell, or both.

### Statistical analyses

Statistical analyses were carried out with Prism 9 (GraphPad Software), using an alpha level of 0.05. All data are presented as the mean ± SEM. Data were tested for normality using the Shapiro–Wilk test. For normally distributed measures, we used an unpaired, two-tailed $t$ test to estimate differences between the two groups. For non-normally distributed data, we used the Mann–Whitney, two-tailed test, whereas when the variances of the groups were not comparable, Welch's test was used. To analyze the gender effect between LMC- and HMC-reared pups, a two-way ANOVA test was used and Tukey's test was used for multiple comparisons. To evaluate litter effect within HMC- and LMC-raised pups, one-way ANOVA test was used for normally distributed data and Tukey's test for multiple comparisons. For non-normally distributed data, Welch's ANOVA test was used and Dunnett's T3 test was used for multiple comparisons. For correlation analyses between either LMC and HMC mother or LMC- and HMC-reared pups at baseline conditions, all possible pairwise correlations were determined by computing Pearson's correlation coefficients. When analyzing multiple correlations simultaneously (correlation matrices), Pearson's coefficients were computed across all numerical variables. $P$-values were corrected using the Benjamini–Hochberg false discovery rate (FDR) procedure. Each complete set of correlations was plotted in color-coded correlation matrices using RStudio. Principal component analyses (PCA) consisted of a series of steps: data selection, scaling, and PCA. For maternal behavior, we selected 17 total measures (in the nest, passive nursing, arched-back nursing, contacting pups, licking or grooming pups, manipulating nest bedding, out of the nest, eating, drinking, self-grooming, rearing, resting away from the litter, pups' retrieval, time spent in the dark chamber during the LDT, DCX fractional volume in DG, DCX fractional volume in OB, and number of pups). For offspring analysis, we selected 18 total measures (DCX fractional volume in the DG, Ki67-positive cells in a $mm^3$, time spent in the dark chamber during LDT, anxiety score, total number of calls at PND5, mean duration of calls at PND5, proportion of all simple calls at PND5, proportion of all elaborated calls at PND5, mean peak frequency at PND5, mean peak amplitude at PND5, total number of calls at PND9, mean duration of calls at PND9, proportion of all simple calls at PND9, proportion of all elaborated calls at PND9, mean peak frequency at PND9, mean peak amplitude at PND9, and occurrence in the nest by the mothers). All measures were standardized using a $z$-score ($z$ = [data point – group mean]/

SD) to account for different units across tests. Principal component analysis: We performed PCA using the prcomp() function in RStudio and calculated the amount of variance explained by each PC using a Scree plot and the loading distribution of principal components using the coefficient outputs from PCA.

## Data Availability

Source data are available as supplementary data and on Zenodo ([77]).

## Supplementary Information

## Acknowledgements

The authors wish to thank Fulvio Magara and Benjamin Boury-Jamot for help and advice with the behavioral experiments. We would also like to thank the caretakers from the Centre d'Etude Comportementale (CEC). This project was funded by the Swiss National Science Foundation (Grant Number 310030_201015).

### Author Contributions

F Grieco: data curation, formal analysis, investigation, methodology, and writing—original draft.
P van Gelderen: investigation.
M Ali: investigation.
T Larrieu: conceptualization, supervision, investigation, methodology, and writing—original draft.
N Toni: conceptualization, supervision, funding acquisition, methodology, project administration, and writing—review and editing.

### Conflict of Interest Statement

The authors declare that they have no conflict of interest.

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
