## [Reviewer comments · Life Science Alliance]

Natural variations in maternal behavior shape anxiety and hippocampal neurogenesis in offspring

Nicolas Toni, Fabio Grieco, Thomas Larrieu, Pauline van Gelderen, and Maryam Ali

DOI: <https://doi.org/10.26508/lsa.202503419>

Corresponding author(s): Nicolas Toni, University of Lausanne and Thomas Larrieu, Lausanne University/Lausanne University Hospital

Review Timeline:

Submission Date:	2025-06-12
Editorial Decision:	2025-09-12
Revision Received:	2025-12-16
Editorial Decision:	2026-01-22
Revision Received:	2026-02-06
Accepted:	2026-02-11

Scientific Editor: Sarita Hebbar

Transaction Report:

September 12, 2025

Re: Life Science Alliance manuscript #LSA-2025-03419-T

Prof. Nicolas Toni
Uni of Lausanne
Psychiatry
Center for Psychiatric Neurosciences
Prilly 1008
Switzerland

Dear Prof. Toni,

Thank you for submitting your manuscript entitled "Natural variations in maternal behavior in inbred mice shape trait anxiety and hippocampal neurogenesis in offspring." to Life Science Alliance. After a delay beyond our control due to reviewer unavailability, we now have two reviewers' reports for your study. We acknowledge your patience in this regards.

The reviewers' comments on the submitted manuscript are appended to this letter. As you will note both reviewers found the study interesting and of potential significance to the field. However, they have also raised several technical concerns that must be addressed for publication at LSA.

Both reviewers raise a point on litter size; we agree that you must provide the relevant information and discuss results in the context of litter sizes. Reviewer 2 has raised an important point on DCX volume as a readout of adult born immature neurons. We concur that you must resolve this point with the requested information and/or data. Finally the suggestions from Reviewer 1 on reanalyses of data taking into account relative occurrence for ABN/LG (quality of maternal behaviour) are important. We agree that they must be included in the revised version.

We invite you to submit a revised manuscript. When submitting the revision, please include a letter addressing the reviewers' comments point by point. While a rebuttal must respond to all points in some form, additional experiments to resolve these points, other than indicated above, are not required.

Thank you for this interesting contribution to Life Science Alliance. We are looking forward to receiving your revised manuscript.

Sincerely,

Sarita Hebbar, PhD
Scientific Editor
Life Science Alliance
<http://www.lsjournal.org>

- A letter addressing the reviewers' comments point by point.
- An editable version of the final text (.DOC or .DOCX) is needed for copyediting (no PDFs).
- High-resolution figure, supplementary figure and video files uploaded as individual files: See our detailed guidelines for

preparing your production-ready images, <https://www.life-science-alliance.org/authors>

B. MANUSCRIPT ORGANIZATION AND FORMATTING:

Reviewer #1 (Comments to the Authors (Required)):

The manuscript "Natural variations in maternal behavior in inbred mice shape trait anxiety and hippocampal neurogenesis in offspring" by Grieco et al. reports measures of behavior and neurogenesis in offsprings and correlate those with maternal behavior.

The authors identify low and high maternal care groups (LMC vs HMC) based on the occurrence in the nest (ON) and report decreased neurogenesis in the hippocampus and olfactory bulb of LMC dams. They measure neonatal behavior (anxiety related and other motor related as control) and neurogenesis in offsprings' hippocampus.

Although the questions raised by the manuscript are interesting and the set of experiments presented provides interesting results, some methodological and conceptual flaws (apparently) need to be addressed or clarified.

Major comments

- The litter size is drastically different between LMC dams (7 dams, 42 pups, i.e. 6 pups per dam) and HMN dams (11 dams, 24 pups, i.e. ~2.2 pups per dam). You showed in Fig 2E that the ON is strongly correlated with the number of pups.
 - o Dams might need to eat/drink much more when feeding more pups. Can you show the linear regression litter size ~ ON (correlation showed in Fig 2E), as well as litter size ~ eating/drinking?
 - o About the correlation between ON and ABN/LG: have you correlated the absolute occurrence or the relative occurrence (i.e. ABN/LG normalised by the ON vs ON?). Of course, if you look at the absolute values, they will be correlated, as a dam who spends a lot of time outside the nest has less time for any maternal care. However once they are on the nest, they could display (relatively) a lot of ABN/LG compared to passive nursing. Otherwise, you cannot say much about the "quality" of the maternal behavior (p7 l. 210)
 - o Methods: in Animals, the number of animals is missing (dams and pups).
 - Did you check for milk spots before the USVs calls? If some pups were hungry and some other not, it might change the nature of the calls.
 - P7, l285 - P8, l240-248: Observations are only correlations, there is no causality between maternal behaviour observations and neonates behavior and neurogenesis.
 - o "these findings suggest that reduced maternal care adversely affects hippocampal neurogenesis in weaned offspring"
 - o "suggesting that low maternal care stimulates the development of inter-individual differences in isolation induced anxiety related behaviour"
 - o "confirming that low maternal care stimulates the development of emotional individuality in isolation-induced anxiety related behavior in pups at early stage"
- >>> As mentioned above, the strong differences in litter size can have multiple consequences, related to the available amount of food first, leading differences in early metabolism.
- >>> Are there dams with a similar litter size but different behavior?

- Behavioural observations: one person who recorded behaviour? Blinded to maternal care? Recorded videos? None of this is specified in the methods.

Other comments

Statistical analyses. Multiple behavioral and molecular tests were carried - how did you correct for multiple comparisons?

Fig 1D: what measure is shown? Is that an average measure of the retrieving multiple pups in the litter (in that case each point has error bars), or have you just run this measure on 1 pup per litter? In methods, specify the number of animals tested.

Fig 1E: how did you measure milk spot? Did you look at the milk spots at the same time of the day for all pups? Methods: nothing is written and you just write in the main text that you checked for milk spots at PND5/6.

Intro could be shortened.

In humans, on the contrary to rodents, fathers also play a role ([https://www.cell.com/trends/neurosciences/fulltext/S0166-2236\(19\)30089-X](https://www.cell.com/trends/neurosciences/fulltext/S0166-2236(19)30089-X)). Although many things are conserved between species, both sexes can display parenting behavior in humans. Careful with quick parallels with rodents.

P3: I85 "liking" >> "licking"

P4: I198 check reference (20 instead of 18?). In ref 20 they seem to say that it is not so clear that neurogenesis is linked to affective behaviour.

P13: I399 please refrain from using familiar language, such as "bad" mothers.

Reviewer #2 (Comments to the Authors (Required)):

Toni and colleagues have addressed the relationship between early-life maternal care and neurogenesis, both in dams and offspring. Several studies have reported that Maternal cares change adult neurogenesis in dams and offspring. However, these studies used artificial maternal cares/intervention. Therefore, the study has addressed the importance of natural variation of maternal cares. First, they found that natural variation of maternal care correlates the levels of adult neurogenesis in DG and OB. Second, they found that natural variation of maternal care correlated with the levels of USVs and adult neurogenesis in juvenile mice. Lastly, they reported that the difference of neurogenesis in DG already exist in P9, indicating that the first week of experience is crucial to set the level of neurogenesis/proliferation in DG, which could have long-term impact on neurogenesis/mood regulation. The experiment was designed well, and the results are clearly presented. There are several technical points to be clarified, but the manuscript is beneficial for the field, and I support to accept this paper after minor revision. If the indicated points are addressed, there is no need to be assessed again.

Minor point 1: The authors used DCX volume to estimate the number of adult-born immature neurons. However, it is unclear whether the volume of the DCX+ area correlates perfectly with the number of DCX+ cells, and this has not been verified or indicated in this manuscript. At the same time, it is unclear how the DCX signal is processed (normalisation, thresholding, etc.). Please clarify this point.

-Minor point 2: How many pups/offspring are used from each litter for histological studies? Are they randomly pick from each litter with the same number of pups. It's not clear whether the contribution from each litter is the same. Also the study basically investigate the variation between litters (with the levels of maternal care), while each data point in histological assessment derived from individual mice. It would be important to show which data point comes from the same litter in the histological data. Then, one could check whether only a few dominating litters is contributing to the changes of mean of AN etc.

REVIEWERS' COMMENTS

Reviewer #1

The manuscript "Natural variations in maternal behavior in inbred mice shape trait anxiety and hippocampal neurogenesis in offspring" by Grieco et al. reports measures of behavior and neurogenesis in offsprings and correlate those with maternal behavior.

The authors identify low and high maternal care groups (LMC vs HMC) based on the occurrence in the nest (ON) and report decreased neurogenesis in the hippocampus and olfactory bulb of LMC dams. They measure neonatal behavior (anxiety related and other motor related as control) and neurogenesis in offsprings' hippocampus.

Although the questions raised by the manuscript are interesting and the set of experiments presented provides interesting results, some methodological and conceptual flaws (apparently) need to be addressed or clarified.

We thank this reviewer for these comments. Please find below a point-by-point response to each of this reviewer's questions. Modifications in the revised version of the manuscript are highlighted in red, to facilitate their tracking.

Major comments

1. • The litter size is drastically different between LMC dams (7 dams, 42 pups, i.e. 6 pups per dam) and HMN dams (11 dams, 24 pups, i.e. ~2.2 pups per dam). You showed in Fig 2E that the ON is strongly correlated with the number of pups.
o Dams might need to eat/drink much more when feeding more pups. Can you show the linear regression litter size ~ ON (correlation showed in Fig 2E), as well as litter size ~ eating/drinking?

In the revised version of the manuscript, we are now showing the linear regressions between litter size and occurrence in the nest, as well as between litter size and eating/drinking (Figure S1C, D).

2. About the correlation between ON and ABN/LG: have you correlated the absolute occurrence or the relative occurrence (i.e. ABN/LG normalised by the ON vs ON?). Of course, if you look at the absolute values, they will be correlated, as a dam who spends a lot of time outside the nest has less time for any maternal care. However once they are on the nest, they

could display (relatively) a lot of ABN/LG compared to passive nursing. Otherwise, you cannot say much about the "quality" of the maternal behavior (p7 l. 210)

The correlation between ON and ABN/LG that was shown in Figure 1E correlated absolute values. In the revised version of the manuscript, we are showing this correlation in Figure 1C. In addition, we are also showing the correlation between ON and ANB/LG normalized by ON (Fig. S1B). Interestingly, this new analysis shows a negative correlation, suggesting that mothers that spent less time in the nest spent proportionally more time in ABN/LG posture. This is now commented in the results section, page 5, paragraph 1.

3. o Methods: in Animals, the number of animals is missing (dams and pups).

We have added the number of dams and pups for each analysis in the materials and methods section.

4. • Did you check for milk spots before the USVs calls? If some pups were hungry and some other not, it might change the nature of the calls.

All pups were checked for the presence of a milk spot before the recording of USV calls and some pups did not display a milk spot before the USV recording. This is now mentioned in the materials and methods section "Ultrasonic vocalization (USV) recording". However, as shown in Figure S2F and G, the presence or absence of a milk spot did not influence the number of calls or the duration of calls. This is now commented in the results section of page 6, first paragraph.

5. • P7, l285 - P8, l240-248: Observations are only correlations, there is no causality between maternal behaviour observations and neonates behavior and neurogenesis.
o "these findings suggest that reduced maternal care aversely affects hippocampal neurogenesis in weaned offspring"
o "suggesting that low maternal care stimulates the development of inter-individual differences in isolation induced anxiety related behaviour"
o "confirming that low maternal care stimulates the development of emotional individuality in isolation-induced anxiety related behavior in pups at early stage"
>>> As mentioned above, the strong differences in litter size can have multiple consequences, related to the available amount of food first, leading differences in early metabolism.
>>> Are there dams with a similar litter size but different behavior?

These sentences and sentences in the discussion have been modified to reflect the association between maternal care and pups' behavior, rather than infer causality. The modifications can be seen in the results section and in the discussion. These changes can also be seen in the titles of the subsections of the results section, in page 7 first paragraph,

page 8 first paragraph, and page 9. We have also changed the title of Fig. S3 and sentences in the discussion.

As shown in Fig.S1D and E in the revised version of the manuscript, there is a correlation between litter size and occurrence in the nest or eating and drinking. This supports the idea that litter size may influence maternal behavior and pups' behavior. However, as can be appreciated from these regressions, dams with the same number of pups still display natural variations in occurrence in the nest. These results suggest that both litter size and natural variations in innate maternal behavior play a role in the expression of maternal behavior. This is now mentioned in the results section (page 6, first paragraph) and discussion (paragraph 6).

6. • Behavioural observations: one person who recorded behaviour? Blinded to maternal care? Recorded videos? None of this is specified in the methods.

All behavioral assessments were performed by a single experimenter, blinded to the maternal care received by the pups. This is now specified in the methods section.

Other comments

7. Statistical analyses. Multiple behavioral and molecular tests were carried - how did you correct for multiple comparisons?

For Supplementary Figure 4, we used a Tukey's multiple comparisons test to correct for multiple comparisons. This is now indicated in the figure legend. For the correlation matrix in Figures 2, 5 and 6, we created new matrices in which the p-values were corrected using the Benjamini-Hochberg False Discovery Rate (FDR) procedure. This is now mentioned in the figure legends and the materials and methods section.

8. Fig 1D: what measure is shown? Is that an average measure of the retrieving multiple pups in the litter (in that case each point has error bars), or have you just run this measure on 1 pup per litter? In methods, specify the number of animals tested.

Each data point represents the average time a dam took to retrieve her offsprings during the test. Due to maternal habituation to pup retrieval, the test was performed using a maximum of 4 pups per litter, when at least 4 pups were available. This is now specified in the figure legends and in the materials and methods section.

9. Fig 1E: how did you measure milk spot? Did you look at the milk spots at the same time of the day for all pups? Methods: nothing is written and you just write in the main text that you checked for milk spots at PND5/6.

The presence of the milk spot was assessed every morning between PND 5 and 6. This is now described in the materials and methods section.

10. Intro could be shortened.

We have shortened the introduction

11. In humans, on the contrary to rodents, fathers also play a role ([https://www.cell.com/trends/neurosciences/fulltext/S0166-2236\(19\)30089-X](https://www.cell.com/trends/neurosciences/fulltext/S0166-2236(19)30089-X)). Although many things are conserved between species, both sexes can display parenting behavior in humans. Careful with quick parallels with rodents.

We have replaced “maternal care” by “parental care” in the human paragraph of the introduction, as well as inserted this citation.

12. P3: l85 "liking" >> "licking"

This is now corrected

13. P4: l198 check reference (20 instead of 18?). In ref 20 they seem to say that it is not so clear that neurogenesis is linked to affective behaviour.

We have shortened the introduction and moved the paragraph about adult neurogenesis in the result section (Page 5, second paragraph). We have also updated the references on this paragraph.

14. P13: l399 please refrain from using familiar language, such as "bad" mothers.
This is now corrected

Reviewer #2:

Toni and colleagues have addressed the relationship between early-life maternal care and neurogenesis, both in dams and offspring. Several studies have reported that Maternal cares change adult neurogenesis in dams and offspring. However, these studies used artificial maternal cares/intervention. Therefore, the study has addressed the importance of natural variation of maternal cares. First, they found that natural variation of maternal care correlates the levels of adult neurogenesis in DG and OB. Second, they found that natural variation of maternal care correlated with the levels of USVs and adult neurogenesis in juvenile mice. Lastly, they reported that the difference of neurogenesis in DG already exist inP9, indicating that the first week of experience is crucial to set the level of neurogenesis/proliferation in DG, which could have long-term impact on neurogenesis/mood regulation. The experiment was designed well, and the results are clearly presented. There are several technical points to be clarified, but the manuscript is beneficial for the field, and I support to accept this paper after minor revision. If the indicated points are addressed, there is no need to be assessed again.

We thank this reviewer for these comments. Please find below a point-by-point response to each of this reviewer's questions. Modifications in the revised version of the manuscript are highlighted in red, to facilitate their tracking.

1. Minor point 1: The authors used DCX volume to estimate the number of adult-born immature neurons. However, it is unclear whether the volume of the DCX+ area correlates perfectly with the number of DCX+ cells, and this has not been verified or indicated in this manuscript. At the same time, it is unclear how the DCX signal is processed (normalisation, thresholding, etc.). Please clarify this point.

As immature neurons express DCX in the soma and the proximal dendrites, the fractional volume can be used as an index that represents a combination of cell number and the morphological complexity of immature neurons. This, and the method to extract this fractional volume, is now clarified in the methods section (Paragraph "Immunohistochemistry")

2. -Minor point 2: How many pups/offspring are used from each litter for histological studies? Are they randomly pick from each litter with the same number of pus. Its not clear whether the contribution from each litter is the same. Also the study basically investigate the variation between litters(withthe levels of maternal care), while each data point in histological assessment derived from individual mice. It would be important to show which data point comes from the same litter in the histological data. Then, one could check whether only a few dominating litters is contributing to the changes of mean of AN etc.

Brains from 87 pups were randomly and blindly selected across litters for histology. To avoid over-representation of individual litters, a comparable number of samples were blindly selected from each litter, irrespective of the maternal care condition received. This is now specified in the materials and methods section of the revised version of the manuscript (Paragraph “Immunohistochemistry”).

In addition, we are now providing a supplementary figure (Fig. S5), displaying the anxiety scores and histology assessments by litter. This enables the visualization inter-litter variability while maintaining the visibility of the histograms of the main figures.

January 22, 2026

RE: Life Science Alliance Manuscript #LSA-2025-03419-TR

Prof. Nicolas Toni
University of Lausanne
Psychiatry
Center for Psychiatric Neurosciences
Prilly 1008
Switzerland

Dear Dr. Toni,

Thank you for submitting your revised manuscript entitled "Natural variations in maternal behavior shape anxiety and hippocampal neurogenesis in offspring". Your revised manuscript was evaluated by both the original reviewers. As you will note, your revisions addressed most of their concerns. However Reviewer 2 notes that their original concern on the correlation between the number of DCX+ cells and volume fraction remains unaddressed.

We noted your addition on page 19 of the methods section of the revised ms to clarify the metric (fractional volume), namely, "As immature neurons express DCX in the soma and the proximal dendrites, the fractional volume can be used as an index comprising cell number and the morphological complexity of immature neurons, as the structural elaboration of these cells reflects their progressive integration into the circuit". But we agree with the reviewer that a clearer explanation is required and the associated limitation with this metric must be explicitly stated.

Overall, we would be happy to publish your paper in Life Science Alliance pending resolution of the above point and final revisions necessary to meet our formatting guidelines.

MANUSCRIPT ORGANIZATION AND FORMATTING:

To avoid unnecessary delays in the acceptance and publication of your paper, please read the following information carefully. Full guidelines are available on our Instructions for Authors page, <https://www.life-science-alliance.org/authors>

- Modify the figure legend for Figure 2E to define the abbreviations as described in the corresponding methods/results section. Please use consistent abbreviations and clearly define if not done so previously in the manuscript text.
- For the legends of Figure 2C and Figure 2D, please describe the arrowheads, and state the size of scale bar in the insets.
- In the methods section, please provide details on objective type and N.A. to complete the description under 'Immunohistochemistry'.
- Please include relevant statements if animal-based experiments were performed in accordance with relevant guidelines and regulation. Please refer to our section on 'Using living organisms & animal welfare' on our website (<https://www.life-science-alliance.org/editorial-policies#animals>).
- Please upload your main manuscript text as an editable doc file.
- The titles in both the system and the manuscript file must be consistent with each other.
- Please be sure that the authorship listing and order are correct.
- Please consult our manuscript preparation guidelines <https://www.life-science-alliance.org/manuscript-prep> and make sure your manuscript sections are in the correct order.
- Please move your main and supplementary figure legends to the main manuscript text after the references section.
- Please use "References" instead of "Bibliography" for the label.
- We encourage you to revise the figure legends for Figure S2 such that the figure panels are introduced in alphabetical order and match the figure.
- There are call-outs for figure S6C-F, and this figure has panels from A-C. Please correct accordingly.
- Please add call-outs for figures S1A and S5A-F.
- Please be sure that the authorship listing and order is correct.
- Please add the X and Bluesky handles of your host institute/organization, as well as your own and/or one of the authors, in our system.

It is Life Science Alliance policy that if requested, original data images must be made available to the editors. Failure to provide

original images upon request will result in unavoidable delays in publication. Please ensure that you have access to all original data images prior to final submission.

LSA encourages authors to provide a 30-60 second video where the study is briefly explained. We will use these videos on social media to promote the published paper and the presenting author (for examples, see <https://docs.google.com/document/d/1-UWCfbE4pGcDdcgzcmiuJl2XMBJnxKYeqRvLLrLSo8s/edit?usp=sharing>). Corresponding or first-authors are welcome to submit the video. Please submit only one video per manuscript. The video can be emailed to contact@life-science-alliance.org

FINAL FILES:

The following items are required for acceptance.

The license to publish form must be signed before your manuscript can be sent to production. A link to the license to publish form will be available to the corresponding author only. Please take a moment to check your funder requirements.

Thank you for your attention to these final processing requirements. Please revise and format the manuscript and upload materials as soon as you are able.

Thank you for this interesting contribution to the literature. We look forward to publishing your paper in Life Science Alliance.

Sincerely,

Sarita Hebbar, PhD
Scientific Editor
Life Science Alliance
<http://www.lsajournal.org>

Reviewer #1 (Comments to the Authors (Required)):

The authors have addressed my comments, I have no further comment.

Reviewer #2 (Comments to the Authors (Required)):

I raised two minor points. The Point 2 was resolved. However, the point 1 has not been resolved. In the manuscript, the author claimed, 'As revealed by a reduction in the number of cells expressing the immature neuronal marker Doublecortin'. They assumed that the fractional volume covered by the DCX signal could represent a combination of cell numbers and morphological complexity. Based on this, it is also possible that only morphological changes are reflected in this index. In order to claim a change in the number of DCX+ cells, the authors should quantify and verify the correlation between the number of DCX+ cells and the volume fraction once.

February 11, 2026

RE: Life Science Alliance Manuscript #LSA-2025-03419-TRR

Prof. Nicolas Toni
University of Lausanne
Psychiatry
Center for Psychiatric Neurosciences
Prilly 1008
Switzerland

Dear Dr. Toni,

Thank you for submitting your Research Article entitled "Natural variations in maternal behavior shape anxiety and hippocampal neurogenesis in offspring". It is a pleasure to let you know that your manuscript is now accepted for publication in Life Science Alliance. Congratulations on this interesting work.

Your manuscript will now progress through copyediting and proofing. At the proofing stage, please define abbreviated terms in Fig 2E legend. We suggest: "E. Pearson's correlation matrix between maternal behavioral parameters, DCX fractional volume in the olfactory bulb (Imm.neurons OB) and hippocampus (Imm.neurons HC), prenatal anxiety, pup number, and eating/drinking."

It is journal policy that authors provide original data upon request.

DISTRIBUTION OF MATERIALS:

Again, congratulations on a very nice paper. I hope you found the review process to be constructive and are pleased with how the manuscript was handled editorially. We look forward to future exciting submissions from your lab.

Sincerely,

Sarita Hebbar, PhD
Scientific Editor
Life Science Alliance
<http://www.lsajournal.org>